# Kummerowia striata extract protects paracetamol-induced liver injury by modulating the S1P/Nrf2/Keap1 pathway

Huanghui Qin[1], Runxiao Chen[1], Youlan Xie[1], Hang Liu[1,2,*], Yubo Xiao[2,*], Lanyu Li[1,*]

**1** Guangxi Key Laboratory of Diabetic Systems Medicine, Guilin Medical University, Guilin, Guangxi, China, **2** Hunan Provincial Key Laboratory for Synthetic Biology of Traditional Chinese Medicine, Hunan University of Medicine, School of Medical Laboratory Science, Hunan University of Medicine, Huaihua, Hunan, China,

* hangliu01@126.com (HL); yuboxiaohn@126.com (YX); 645085995@qq.com (LL)

## Abstract

### Background

Drug-induced liver injury remains a significant challenge in both drug development and clinical applications. Acetaminophen (APAP), a commonly used analgesic and antipyretic, can cause liver damage when overdosed, with APAP-induced liver injury being one of the leading causes of acute liver injury. *Kummerowia striata* (Ks), a leguminous plant, has been reported to contain chemical compounds with anti-inflammatory and antioxidant activities. However, its potential therapeutic effects in liver diseases remain inadequately explored.

### Objective

This study aims to investigate the protective effect of *Kummerowia striata* extract against APAP-induced acute liver injury and further explore its potential molecular mechanisms.

### Methods

C57BL/6J mice were pre-treated with Ks extract by gavage for 3 days. On the second day, treatment was discontinued, and the mice were fasted for 16 hours. After the fasting period, the final dose of Ks extract was administered, followed by an intraperitoneal injection of APAP (300 mg/kg) 1 hour later to establish a drug-induced liver injury model. Tissue samples were collected 24 hours after modeling. To investigate the molecular mechanisms by which Ks extract prevents APAP-induced acute liver injury, network pharmacology, the GEO database, and molecular docking analysis were employed.

**Data availability statement:** All relevant data are within the manuscript and its Supporting information files.

**Funding:** This research was supported by Guangxi Natural Science Foundation Joint Special Project (Guilin Medical University Special Project, 2024JJH130036) and Natural Science Foundation of Hunan Province of China (2023JJ50440). The funders had no role in study design, data collection and analysis, decision to publish, or preparation of the manuscript.

**Competing interests:** The authors have declared that no competing interests exist.

## Results

Ks extract exhibited a significant gender-independent protective effect in preventing APAP-induced acute liver injury in mice. In male mice, Ks extract attenuated the occurrence of acute liver injury by modulating the sphingosine-1-phosphate/ Sphingosine-1-Phosphate Receptor 2/ Sphingosine-1-Phosphate Receptor 4(S1P/S1PR2/S1PR4) signaling pathway and the upstream regulator SPHK1. This, in turn, regulated the binding of Kelch Like ECH Associated Protein 1 (Keap1) to Nuclear factor erythroid 2-related factor 2(Nrf2), alleviating oxidative stress and inflammatory responses.

## Discussion

This study illustrates that Ks exerts a protective effect against APAP-induced acute liver injury by modulating oxidative stress and inflammatory responses through multiple mechanisms. Notably, Ks inhibited Keap1 expression and restored Nrf2 signaling, indicating a potential regulatory role in mitigating mitochondrial oxidative damage and ferroptosis. Furthermore, Ks downregulated SPHK1 expression and reduced levels of the bioactive sphingolipid S1P, as well as its receptors S1PR2 and S1PR4, unveiling a previously unreported involvement of the SPHK1/S1P/S1PR2/4 axis in liver injury. Interestingly, sex-based differences in APAP hepatotoxicity were observed, with female mice exhibiting lower susceptibility, yet still responding to Ks treatment. These findings not only validate the hepatoprotective potential of Ks but also provide new mechanistic insights involving the Nrf2/Keap1 and S1P/S1PR2/S1PR4 signaling pathways, thereby laying a foundation for its development as a candidate therapeutic agent for acute liver injury.

## 1. Introduction

The liver is the primary metabolic organ in the body, responsible for drug metabolism and biotransformation. It is also the main target organ for injury caused by drugs or chemicals. Drug-induced liver injury (DILI) represents a significant challenge in both drug development and clinical applications [1]. Due to the adverse effects of DILI, many drugs are discontinued during clinical trials, and some are withdrawn from the market due to the occurrence of DILI. Acetaminophen (APAP), also known as paracetamol, is widely used in clinical practice for pain relief and fever reduction. However, excessive use of APAP is one of the most common causes of drug-induced liver injury in clinical settings [2]. According to statistics from the U.S. FDA in 2017, APAPis responsible for 46% of all cases of acute liver failure, with at least 500 deaths annually attributed to APAP overdose [3,4]. Globally, there is a wide range of Western pharmaceuticals and compound traditional Chinese medicine formulations containing acetaminophen, particularly those used for cold treatments. Most of these medications are available over-the-counter, making them widely accessible and easily purchasable in the market [5–7]. Therefore,

it is not uncommon for patients to experience liver injury due to the concurrent use of multiple medications containing APAP. APAP is primarily metabolized in the liver through two pathways, with 70–80% being conjugated with glucuronic acid and sulfate. However, during overdose, APAP is metabolized via the cytochrome P450 enzyme system (mainly CYP2E1, CYP3A4, and CYP1A2), leading to the generation of a toxic metabolite, N-acetyl-p-benzoquinone imine (NAPQI). When NAPQI is produced in excess, it depletes glutathione (GSH) and covalently binds to liver cell proteins and DNA, thereby inducing mitochondrial oxidative stress and disrupting lipid metabolism. Ultimately, this results in severe hepatotoxicity and hepatocyte necrosis [8,9].

Sphingolipid signaling has been recognized as a critical mediator of inflammation and cancer progression. As bioactive second messengers, sphingolipids participate in key intercellular communication pathways, with their metabolic regulation being intimately associated with inflammatory processes and oxidative stress responses [10]. Previously, research on sphingolipids in liver diseases has primarily focused on areas such as liver cancer, fatty liver disease (including non-alcoholic fatty liver disease), liver fibrosis, and liver regeneration. However, studies on their role in liver injury remain relatively scarce [10,11]. S1P is a structural, metabolic, and bioactive lipid involved in the regulation of various physiological responses, including cell growth, transformation, migration, and cell death [12]. Multiple studies have demonstrated the crucial role of S1P and its transmembrane receptors (S1PR) in both the regulation of inflammatory responses [13]. Studies have indicated that liver injury is closely associated with abnormalities in sphingolipid metabolites. Modulating the SPHK1/S1P signaling pathway can alleviate liver injury induced by LPS/D-GalN and ischemia [14,15]. Additionally, Ceramide Kinase/ Ceramide-1-Phosphate (CERK/C1P) may influence acute liver injury induced by carbon tetrachloride by modulating the Nrf2/Keap1 signaling pathway [16]. In the context of drug-induced liver injury (DILI), studies have shown that the deficiency of SPHK1 can protect mice from APAP-induced DILI. However, other research indicates that stress-induced upregulation of SPHK1 can inhibit NLRP3 inflammasome activation and hepatocyte pyroptosis, thereby alleviating APAP-induced DILI [17]. Therefore, we believe that the role of SPHK1 in DILI and its underlying mechanisms require further investigation, and that drugs targeting SPHK1 are yet to be developed.

SPHK1 and Sphingosine kinases2 (SPHK2) are two isoenzymes responsible for regulating the synthesis of S1P. Both SPHK1 and SPHK2 are abundantly expressed in the liver, although SPHK1 is predominantly expressed in cardiomyocytes. S1P, a bioactive sphingolipid metabolite and one of the most extensively studied members of the sphingolipid family, has been shown to play a crucial role in the pathogenesis of inflammation and oxidative stress [18]. Targeted modulation of S1P signaling has emerged as a promising therapeutic strategy for various liver diseases, including hepatic steatosis, alcoholic liver injury, liver fibrosis, cirrhosis, and hepatocellular carcinoma. However, investigations into the role of S1P in DILI remains scarce. A comprehensive review of the literature revealed a lack of studies focusing on S1P and its five known receptors (S1PR1–5) in the context of DILI [19]. Notably, SPHK1, which regulates S1P levels, has been implicated in APAP-induced DILI. Nevertheless, the regulatory effects and underlying mechanisms of SPHK1 remain highly controversial. Motivated by this gap in the literature, we aimed to elucidate the role of SPHK1 regulation and the impact of altered S1P levels on DILI, with a specific focus on the SPHK1/S1P signaling axis [14,15].

Nrf2 signaling pathway is a critical regulatory transcription factor in oxidative stress, inflammation, and ferroptosis [20]. Studies have shown that in the cytoplasm, Keap1frequently binds to Nrf2 and targets it for continuous degradation through the ubiquitin-proteasome pathway [21]. Upon stimulation, Nrf2 is released and translocates to the nucleus. Additionally, it can disrupt the Keap1-Nrf2 complex through competitive binding, leading to the activation of the Nrf2 signaling pathway. Liver injury is often associated with oxidative stress or ferroptosis resulting from lipid peroxidation. Several drugs have been shown to alleviate oxidative stress in vitro by modulating Nrf2 and its downstream factors (such as NQO1, HO-1), thereby preventing liver injury [22–24]. Consequently, regulation of Nrf2 has been widely recognized as a viable strategy for mitigating oxidative stress and preventing drug-induced liver injury (DILI).

*Kummerowia striata* (Ks), commonly known as "chicken-eye grass," was first recorded in the Jiu Huang Ben Cao(Compendium of Emergency Food for Famine Relief) and has been included in the medicinal literature of the ethnic minorities

of Guangxi, Zhuang Yao Xue (Zhuang Medicine). Ks is an important ingredient in the Miao medicinal formula Qi Bai Ma Niu Yang Ji Tang (Seven White Horse, Ox, Sheep, and Chicken Soup), which has demonstrated significant therapeutic effects, particularly for viral hepatitis and syndromes caused by liver disease and pathogenic factors. In this formula, Ks is considered effective in clearing liver heat, thus playing an important role in traditional medicine. Western medicine recognizes that drug-induced liver injury is caused by hepatotoxic drugs, often referred to as "liver poisoning." Despite evidence supporting the anti-inflammatory and antioxidant properties of Ks extracts, limited research has been conducted on their efficacy in liver diseases. Thus, the therapeutic potential of this plant for hepatic disorders remains to be fully elucidated. This highlights the critical need for continued research and development of Ks within the context of hepatic injury. Nevertheless, the protective role of Ks and the underlying mechanisms in cases of drug-induced liver injury are not yet fully understood. Nrf2 is a pivotal regulatory factor in managing oxidative stress and inflammatory responses, and the modulation of Nrf2 expression and activity is widely acknowledged as a therapeutic strategy of acute liver injury [25]. Therefore, this study aims to investigate the protective effect of Ks extract on APAP-induced acute liver injury in mice, and to elucidate its mechanisms in improving oxidative stress and inflammatory response through sphingolipid metabolism and the Nrf2 signaling pathway.

## 2. Materials

### 2.1. Animals

SPF-grade healthy male C57BL/6 mice, aged 6–8 weeks and weighing 18–22 g, were purchased from the Animal Experiment Center of Guilin Medical University. The animals were accompanied by the Animal Production License and the Animal Use Certificate. All animal experiments were approved by the Ethics Committee for Animal Experiments of Guilin Medical University.

### 2.2. Drugs

The extract of *Kummerowia striata* was obtained from Xi'an Luruquan Biotechnology Co., Ltd., with a ratio of extract to raw material of 50:1 (Catalog No. LRQ240221−1). The extract was dissolved in distilled water for use. Acetaminophen (purity 99%, AR) was purchased from MCE (Catalog No. HY-66005) and freshly prepared for each experiment. It was dissolved by heating in a 70°C water bath for 10 minutes and then stored at 45°C. The dosage was determined based on preliminary experimental studies.

### 2.3. Reagents

4% Tissue Cell Fixative (4% Paraformaldehyde) (purchased from: Solarbio; Catalog No.: P1110); Alanine Aminotransferase (ALT) Assay Kit (IFCC method) (purchased from: Mindray; Catalog No.: 140124009); Aspartate Aminotransferase (AST) Assay Kit (IFCC method) (purchased from: Mindray; Catalog No.: 140224004); Lactate Dehydrogenase (LDH) Assay Kit (IFCC method) (purchased from: Mindray; Catalog No.: 142723011);Total Superoxide Dismutase (T-SOD) Assay Kit (WST-1 method) (purchased from: Nanjing Jiancheng; Catalog No.: A001-3–2);Malondialdehyde (MDA) Assay Kit;Reduced Glutathione (GSH) Assay Kit (Microplate method) (purchased from: Nanjing Jiancheng; Catalog No.: A006-2–1);Mouse Interleukin-6 (IL-6) ELISA Quantitative Detection Kit (purchased from: Proteintech; Catalog No.: KE10007); Mouse Interleukin-1 Beta (IL-1β) ELISA Kit (purchased from: Elabscience; Catalog No.: E-EL-M0037); Mouse Tumor Necrosis Factor Alpha (TNF-α) ELISA Kit (purchased from: Elabscience; Catalog No.: E-EL-M3063); Mouse Interleukin-10 (IL-10) ELISA Kit (purchased from: BY Abscience; Catalog No.: BY-EM220162); Sphingosine 1-Phosphate (S1P) Assay Kit (purchased from: Echelon Biosciences; Catalog No.: K-1900); Mouse Sphingosine Kinase-2 (Sphk-2) ELISA Quantitative Detection Kit (purchased from: UpingBio; Catalog No.: SYP-M2060); Mouse SPHK1 ELISA Kit (purchased from: UpingBio; Catalog No.: SYP-M2057); Mouse S1PR2 ELISA Kit (purchased from: UpingBio; Catalog No.: SYP-M1458);

Mouse S1PR4 ELISA Kit (purchased from: BY Abscience; Catalog No.: BYHS505577); Mouse Catalase (CAT) ELISA Kit (purchased from: Jinhengnuo; Catalog No.: 231024JN); Mouse Keap1 ELISA Kit (purchased from: WeiaoBio; Catalog No.: EM30387XS); SPHK1 Polyclonal Antibody (purchased from: Proteintech; Catalog No.: 10670–1-AP); NFE2L2 (Nrf2) Recombinant Antibody (purchased from: Proteintech; Catalog No.: 8059–1-RR); Heme Oxygenase 1 Antibody (purchased from: Abways; Catalog No.: CY8761); NQO1 Antibody (purchased from: Abways; Catalog No.: CY6710); Omni-Easy™ Ready-to-use BCA Protein Quantification Kit (purchased from: Yamei; Catalog No.: ZJ102); General Antibody Dilution Buffer (purchased from: Yamei; Catalog No.: PS119); ToloScript All-in-one RT EasyMix for qPCR (One-step Reverse Transcription) (purchased from: Tolobio; Catalog No.: 22107); 2xQ5 SYBR qPCR Master Mix (Universal) (purchased from: Tolobio; Catalog No.: 22206−01).

## 3. Methods

### 3.1. Animal model, grouping, drug administration, and sample collection

Forty-two C57BL/6 mice were randomly divided into six groups, including the blank control group, model group (APAP), positive drug group (N-acetylcysteine, NAC), and three Ks extract dose groups (60 mg/kg, 120 mg/kg, 240 mg/kg), with seven mice in each group. The Ks extract was administered by oral gavage at the respective doses. The positive drug group was given 200 mg/kg N-acetylcysteine (NAC) via oral gavage, while the normal and model groups received distilled water by gavage. All treatments were administered once daily for three consecutive days.

After the last dose was administered, the mice fasted for 16 hours, beginning 5 hours after the final administration of the treatment. Following the fasting period, the final dose was given. One hour after the final dose, the blank control group received an intraperitoneal injection of saline, while the remaining groups received an intraperitoneal injection of 300 mg/kg APAP (dissolved in saline) to induce drug-induced acute liver injury.

Twenty-four hours after intraperitoneal injection, the mice were weighed. Blood was collected via ocular puncture, and the samples were allowed to stand at room temperature for 1 hour. The serum was then separated by centrifugation at 3000 rpm for 15 minutes at 4°C. The supernatant (serum) was collected and stored at −80°C. Mice were euthanized by cervical dislocation, and the liver was dissected. The liver was rinsed with physiological saline, and the surface liquid was blotted dry with filter paper before weighing. The largest lobe of the liver was divided into three portions and fixed in 4% paraformaldehyde for subsequent histopathological analysis. The remaining liver tissue was minced and frozen in liquid nitrogen before being stored at −80°C for further analysis.

### 3.2. UPLC-Q-TOF-MS analysis

Extraction and Analysis of Active Compounds:10 mg of Ks extract was dissolved in 1 mL of 70% methanol aqueous solution to extract the active components. The extracted active compounds were analyzed using a Thermo Fisher Q-Exactive instrument. Instrumental Settings: The electrospray ionization (ESI) source was employed in both positive and negative ion modes (ESI+ and ESI-), and full scan/data-dependent MS/MS (Full MS/ ddMS2) was used for the analysis. The main settings for the source area were as follows: sheath gas flow rate of 9 arb, spray voltage of 3 kV, capillary temperature of 320°C, S-lens voltage of 55 kV, and auxiliary gas heater temperature of 30°C. The MS full scan mode (Full MS) had a scanning range of m/z 100–1200, with a resolution of 70,000. The number of ions entering the C-Trap was 1e6, and the maximum ion injection time was 100 ms.In MS/MS mode (ddMS2), the resolution was set at 35,000, with the peak excitation time ranging from 3 to 9 s. The collision energy was set in a stepped mode at 20 kV, 40 kV, and 60 kV, with a dynamic exclusion time of 8 s.

### 3.3. Measurement of serum ALT, AST, and LDH activities

After thawing the serum stored in section 2.1 on ice, 30 μL of the serum was diluted 10-fold with normal saline. Following the manufacturer's instructions, enzyme activities of ALT, AST, and LDH were measured using a BS-460 fully automated biochemical analyzer for each group.

### 3.4. Liver tissue histopathology

Liver tissue fixed as described in section 2.1 was dehydrated, cleared, and embedded before being stored at room temperature. When required, 4 µm thick sections were cut and placed in an oven at 70°C for 2 hours prior to staining. The sections were then dewaxed using xylene, stained with Hematoxylin-Eosin (HE), and subjected to dehydration through a graded ethanol series and clearing with xylene. After embedding in neutral resin, the specimens were observed under an optical microscope.

### 3.5. Real-time quantitative PCR (RT-qPCR)

Tissues were collected, and RNA was extracted using the TRIzol reagent RNA extraction kit. The concentration and purity of the samples were measured using a UV spectrophotometer. According to the manufacturer's instructions, RNA was reverse transcribed into complementary DNA (cDNA) using the M-MLV Reverse Transcriptase Kit (RNase H-). Primers were designed for different genes, with special attention given to variable regions to avoid primer specificity issues. The mouse GAPDH gene was used as an internal control for RT-qPCR experiments. The target sequences were amplified based on the selected primers and templates, and the accumulation of fluorescence signals during amplification was monitored using fluorescent probes. After the reaction, the relative expression of the target genes was automatically calculated by software using the Ct values obtained from the real-time PCR assay. The relative expression levels were then analyzed using the 2-ΔΔCt method. The specific calculation formulas are as follows: ΔCt = the average Ct value of the target gene of the sample – the corresponding Ct value of the internal reference gene (GAPDH); ΔΔCt = ΔCt of the experimental group - ΔCt of the control group. Relative gene expression levels in each group were then obtained by applying the 2-ΔΔCt method, with results expressed relative to the control group.

### 3.6. ELISA

The 10% liver tissue homogenated or serum prepared as described above was used for ELISA assays. After determining the appropriate dilution ratio through preliminary experiments, the levels of inflammation and sphingolipid metabolism-related molecules in different groups of mice were measured.

### 3.7. Measurement of glutathione (GSH) content

The GSH content in liver tissue homogenates was determined using a GSH detection analysis kit, according to the manufacturer's instructions.

### 3.8. Measurement of Superoxide Dismutase (SOD) and Malondialdehyde (MDA) content

The MDA content in liver tissue homogenates was measured following the instructions provided with the MDA detection kit.

### 3.9. Western blotting

Liver tissue samples from animals were lysed using IP lysis buffer according to the kit instructions to extract protein. The protein concentration was determined using the BCA method. Based on the calculated concentration, pure water, protein solution, and loading buffer were sequentially added to prepare protein samples. After vortexing and centrifugation, the protein samples were boiled for 5 minutes to denature the proteins. The samples were stored at −20°C or −80°C. For samples to be analyzed in the same batch, the protein concentration was adjusted to uniformity with 2% sodium dodecyl sulfate (SDS) solution and mixed with protein loading buffer for Western blot analysis. Proteins were separated by SDS-polyacrylamide gel electrophoresis (PAGE) using 4% and 10% gels. The proteins were transferred to polyvinylidene fluoride (PVDF) membranes under constant current (210 mA). The membranes were blocked with 5% non-fat milk for 2

hours and incubated with primary antibody at 4°C overnight. GAPDH was used as the internal reference. After primary antibody incubation, the membranes were washed and incubated with secondary antibody at room temperature for 2 hours. Following washing, chemiluminescence detection and imaging were performed, and protein band intensity was analyzed by grayscale semi-quantification.

### 3.10. Molecular docking

The 3D protein structure of the target compound was obtained from the Protein Data Bank (PDB). Additionally, the 2D chemical structure of the small molecule was downloaded from PubChem, a comprehensive repository of chemical compounds maintained by the National Center for Biotechnology Information (NCBI) [26]. Molecular docking was performed using AutoDock Vina 1.5.6 [27]. After docking, the conformation with the highest frequency of occurrence and the best binding effect was selected as the final result.

### 3.11. Statistical analysis

Data were summarized using Excel, and statistical analysis and graph generation were performed using GraphPad 8.0.2 (GraphPad Software, Boston, Massachusetts, USA, www.graphpad.com). Pairwise comparisons were conducted using the paired t-test. A $p$-value of $< 0.05$ was considered statistically significant.

## 4. Results

### 4.1. UPLC-Q-TOF-MS analysis

The chemical composition of Ks was analyzed using UPLC-Q-TOF-MS/MS technology. Based on reference standards, retention time, molecular weight fragmentation peaks, and a database, a total of 595 compounds were identified. The chemical composition information of Ks in vitro is shown in Appendix 1 (S3 Fig).

### 4.2. Ks attenuates APAP-induced liver injury

We first investigated the effect of Ks on liver injury in a mouse model (Fig 1A). Under different treatment conditions, the liver appearance in the model group exhibited significant liver enlargement, a pale red and uneven color, with visible punctate hemorrhages and necrotic lesions. The diaphragm tension was increased, and numerous millet-like white spots were observed. In contrast, both low and high-dose Ks treatment groups significantly ameliorated these phenomena (Fig 1C). The liver-to-body weight ratio is an important indicator for assessing liver health in mice. In the APAP model group, the liver-to-body weight ratio was significantly higher than that of the control group. However, following preventive treatment with Ks or NAC, the liver-to-body weight ratio in the treated groups was significantly lower compared to the model group (Fig 1B). Compared to the control group, the serum liver function markers in the APAP group were significantly elevated, indicating the successful establishment of the drug-induced liver injury (DILI) model. In comparison to the APAP group, both low and high-dose Ks treatments significantly reduced ALT, AST, and LDH levels, with the high-dose group showing effects approaching those of the positive control drug (Fig 1D, E, and F).

   To observe the pathological changes in the liver of mice, we prepared tissue sections and performed Hematoxylin and Eosin (HE) staining. The pathological results showed that in the normal group, hepatocytes were radially arranged around the central vein, with clear liver cord structure. No dilation of liver sinusoids was observed, and there was no significant hepatocyte necrosis or degeneration. No inflammatory cell infiltration or fibrotic proliferation was seen in the liver lobules and portal areas (Fig 1G). In the APAP group, the liver sinusoids were significantly dilated with congestion, accompanied by extensive infiltration of inflammatory cells. Cytoplasmic eosinophilia was observed in many hepatocytes, indicating eosinophilic degeneration. Bridging necrosis was visible within the liver lobules, primarily surrounding the central vein, with areas of congestion in some necrotic regions. Chromatin condensation, apoptotic bodies with cytoplasmic shrinkage,

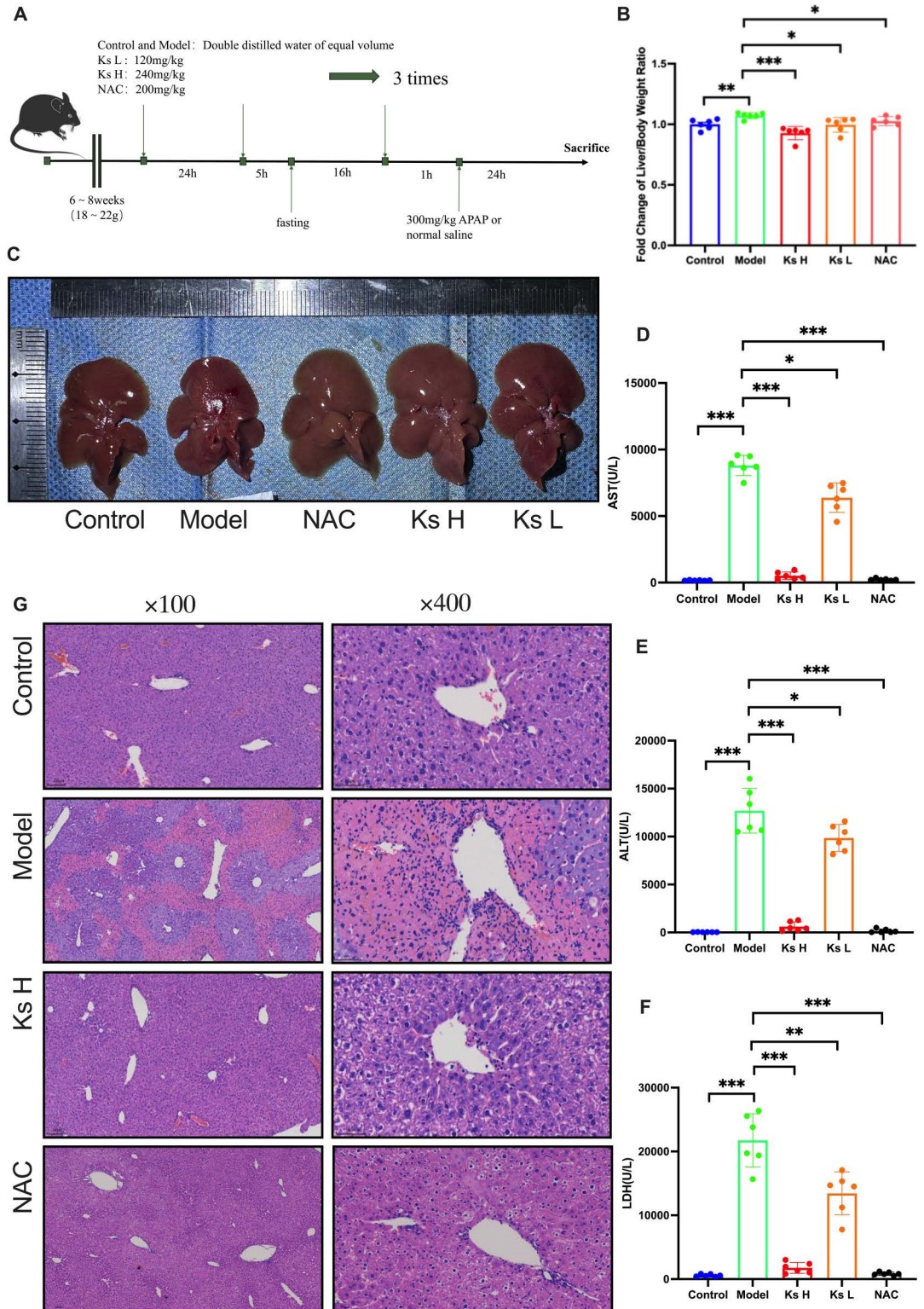

**Fig 1. Ks Attenuates APAP-Induced Liver Injury.** (A) Schematic diagram of the animal experiment. Mice were administered orally for two consecutive days, followed by 16 hours of fasting. One hour after the final administration, mice received either saline or APAP (300 mg/kg, i.p.). Animals were

sacrificed 24 hours later.(B) Liver-to-body weight ratio of the mice.(C) Representative images of mouse livers.(D–F) Serum levels of AST, ALT, and LDH were measured to assess liver function.(G) H&E staining of liver tissue sections.*$P<0.05$, **$P<0.01$, ***$P<0.001$ indicate statistically significant differences compared with the control or model group.

vacuoles, and large granular bilirubin-like deposits were observed at the necrotic margins. The endothelial cell nuclei of the central veins were damaged, showing signs of karyolysis and karyorrhexis. The low-dose group significantly reduced the area of necrosis, while the high-dose Ks group showed well-organized liver cords with no apparent congestion or dilation of the liver sinusoids. Only a small number of inflammatory cells and eosinophilic changes were observed. These results suggest that Ks exhibits significant hepatoprotective effects and can effectively improve liver damage induced by APAP.

#### 4.3. Active targets and enriched pathways of Ks in the prevention of APAP-induced acute liver injury

By integrating research on the active ingredients of *Kummerowia striata* extracts from databases such as Herb, TCMSP, CNKI, and PubMed, we identified 13 major components, including apigenin, apigenin-7-O-glucoside, kaempferol, and luteolin. After inputting the SMILES structures of these components into the PubChem database and subsequently entering the information into the Swiss Target Prediction database, we removed duplicate data and ultimately identified 200 potential targets (Fig 2B). Next, we matched the targets of *Kummerowia striata* with those associated with acute liver injury, resulting in 187 potential targets. Using Cytoscape, we constructed a protein-protein interaction (PPI) network to further identify key genes involved in the therapeutic effects of *Kummerowia striata* on acute liver injury (Fig 2A and C). We performed Gene Ontology (GO) analysis and KEGG pathway enrichment analysis on these potential targets, filtering out irrelevant pathways (Fig 2D and E). In the GO analysis, top terms including plasma membrane, cytosol, zinc ion binding, negative regulation of apoptotic process and so on.KEGG pathway enrichment analysis revealed significant effects on Sphingolipid signaling pathway, Metabolic pathways and so on. We used PubChem to retrieve the Canonical SMILES of 595 compounds identified through mass spectrometry analysis. These compounds were screened based on the criteria of having a "high gastrointestinal absorption score" and fulfilling at least four out of five conditions for drug similarity, marked as "yes." The 101 compounds passing the screening were then imported into the Swiss Target Prediction database for target prediction. Subsequently, gene names were standardized using the Uniprot platform. The predicted drug targets were cross-referenced with disease-associated differential genes from the GSE241511 dataset. Finally, we repeated the same process to predict the potential targets of Ks extract for preventing acute liver injury caused by APAP(S2 Fig).

#### 4.4. Ks can alleviate oxidative stress and inflammation caused by APAP-induced liver injury

IL-1β, IL-6, and TNF-α are well-recognized pro-inflammatory cytokines, while IL-10 is an important anti-inflammatory cytokine that is closely associated with the pathogenesis of various diseases [28–30]. Numerous studies have shown that liver injury can lead to elevated levels of pro-inflammatory factors and suppress the expression of anti-inflammatory factors. Through network pharmacology analysis, we predicted that AKT1, TNF-α, and STAT3 are key targets, with redox enzyme activity being the primary affected pathway. The interactions between these predicted targets and inflammatory factors are closely related.

We quantitatively analyzed the protein levels of IL-1β, IL-6, TNF-α, and IL-10 in liver tissues using ELISA. The results were consistent with our expectations. In the APAP group, the protein levels of IL-1β, IL-6, and TNF-α in liver tissues were significantly higher compared to the Control group, while the protein level of IL-10 was significantly suppressed. However, treatment with the drug significantly reduced the levels of these pro-inflammatory factors and promoted the levels of the anti-inflammatory factor IL-10 (Fig 3E–H).

                                                                                    

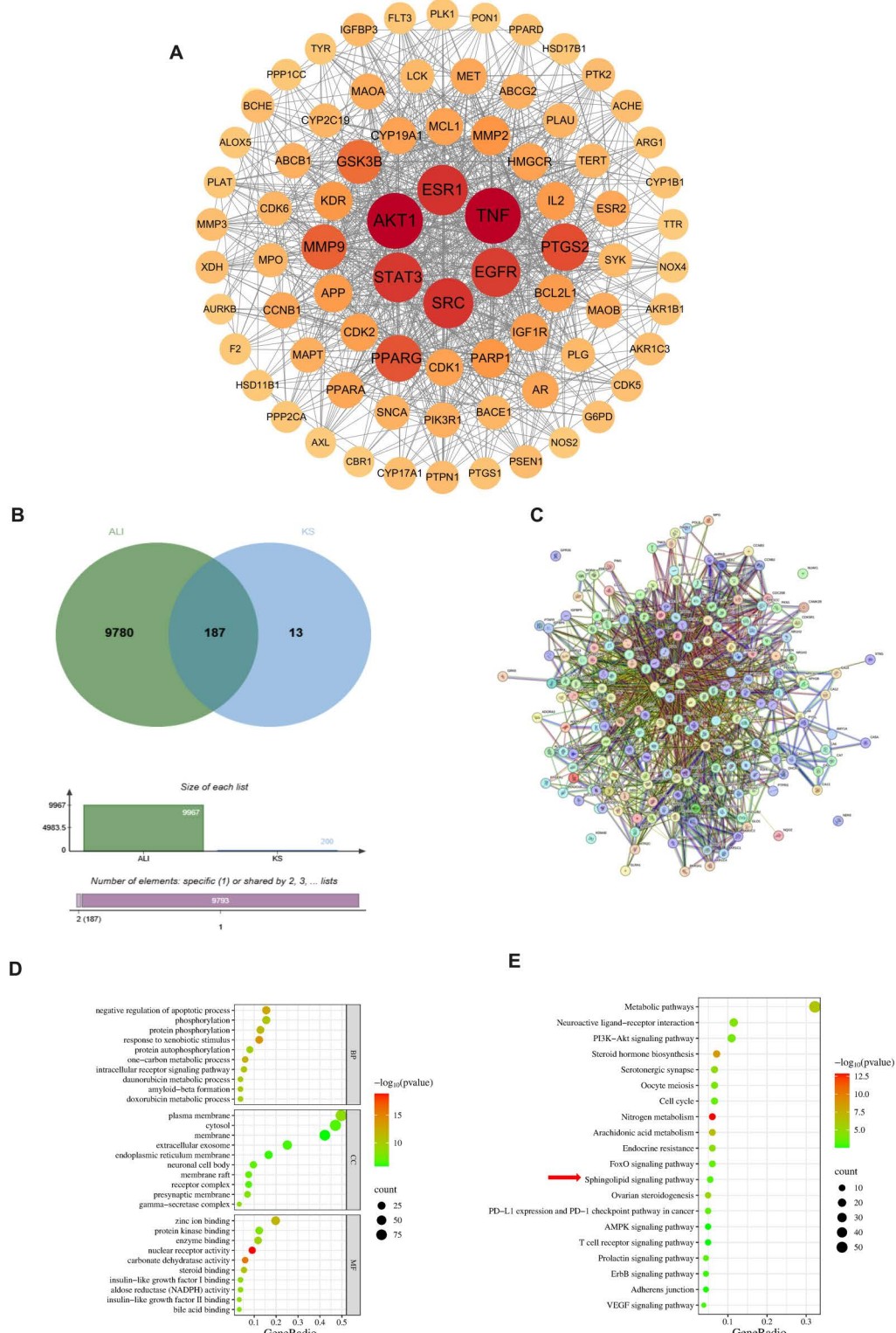

**Fig 2. Active Targets and Enriched Pathways of Ks in the Prevention of APAP-Induced Acute Liver Injury.** (A) Drug Compound Target Pathway-Disease Network. The depth of color represents the strength of targeting effect.(B) Venn Diagram of Targets of Ks Associated with Acute Liver Injury

(ALI).(C) Interaction of Key Genes.(D) GO Enrichment Analysis of Target Genes.(E) KEGG Pathway Analysis of Target Genes. The bubble size indicates the number of genes in the pathway, and the color represents the p-value.(Specific analysis can be found in the original text.).

Subsequently, we selected the blank group, model group, and high-dose group for further study. Using qPCR, we examined the mRNA expression changes of representative inflammatory factors, including IL-1, IL-6, TNF-α, and IL-10. The results showed that, at the mRNA level, the APAP group significantly upregulated the expression of IL-1, IL-6, and TNF-α, while suppressing IL-10 expression. In contrast, the KsH group significantly inhibited the expression of IL-1, IL-6, and TNF-α, while promoting IL-10 expression. The trends in mRNA expression were consistent with the ELISA results (Fig 3A–D).

SOD, MDA, GSH, and CAT are key indicators reflecting the extent of oxidative stress [16,31,32]. To investigate the oxidative stress status in the liver of mice under different treatment conditions, we measured these parameters. The results showed that mice subjected to APAP modeling exhibited significant oxidative stress, which was notably alleviated following treatment (Fig 3IL).

### 4.5.  Ks promotes Nrf2 nuclear translocation and activates the Nrf2/Keap1 pathway

Nrf2 is one of the key regulatory pathways involved in oxidative stress and inflammation, with its regulatory mechanisms encompassing multiple pathways [16,31]. To investigate whether Nrf2 is a potential target of Ks, we assessed the mRNA expression levels of Nrf2 and Keap1 through qPCR. The results showed that, at the gene level, Keap1 expression remained unchanged, whereas Nrf2 expression was significantly reduced in the model group but markedly increased in the high-dose group (Fig 3M and N). Further analysis via Western blot and immunohistochemistry revealed that both Keap1 and Nrf2 protein expressions were upregulated in the model group, but significantly decreased in the high-dose group. While previous studies commonly reported that Keap1 expression did not change in the APAP-induced liver injury model, we observed high expression of Keap1 at the protein level (Figs 3O and P, 4A and B) [16,20,32]. To exclude the possibility of false-positive signals in the WB and IHC experiments, we performed an ELISA to quantitatively analyze the Keap1 levels, and the results were consistent with those of the WB analysis (Fig 3Q). Next, we examined the expression changes of Nrf2 downstream proteins NQO1 and HO-1. qPCR analysis showed that NQO1 expression did not significantly change under different treatments, while HO-1 was highly expressed in the model group and significantly suppressed following drug intervention (Fig 4F and G). WB experiments further confirmed these changes, in line with the Nrf2 expression trend; both NQO1 and HO-1 were upregulated in the model group, but notably decreased after treatment (Fig 4D and E).

### 4.6.  Ks inhibits the metabolism of sphingolipid metabolite S1P and improves APAP-induced acute liver injury

Due to the limited data available on the components of Ks in network pharmacology, we aimed to explore potential drug targets of Ks by analyzing datasets from the GEO public database that matched our experimental protocol. The goal was to identify new disease and drug targets.

Sphingolipids, as second messengers in cells, are involved in the pathogenesis of various diseases, particularly those related to inflammatory responses, fibrosis, and metabolic disorders. However, research on their role in liver injury remains insufficient [33]. Previous studies have shown that Ks exhibits significant anti-inflammatory effects, and literature has reported that IL-10 can act as an upstream regulator of sphingolipid metabolism [30]. In the network pharmacology GO analysis, sphingolipid metabolism was identified as one of the significant pathways. Subsequently, we analyzed the dataset GSE241511 and found that several genes related to sphingolipid metabolism (such as Smpd3, CerS6, sphk1, S1PR3, and S1PR5) exhibited significant alterations following APAP-induced modeling in mice, with the most prominent

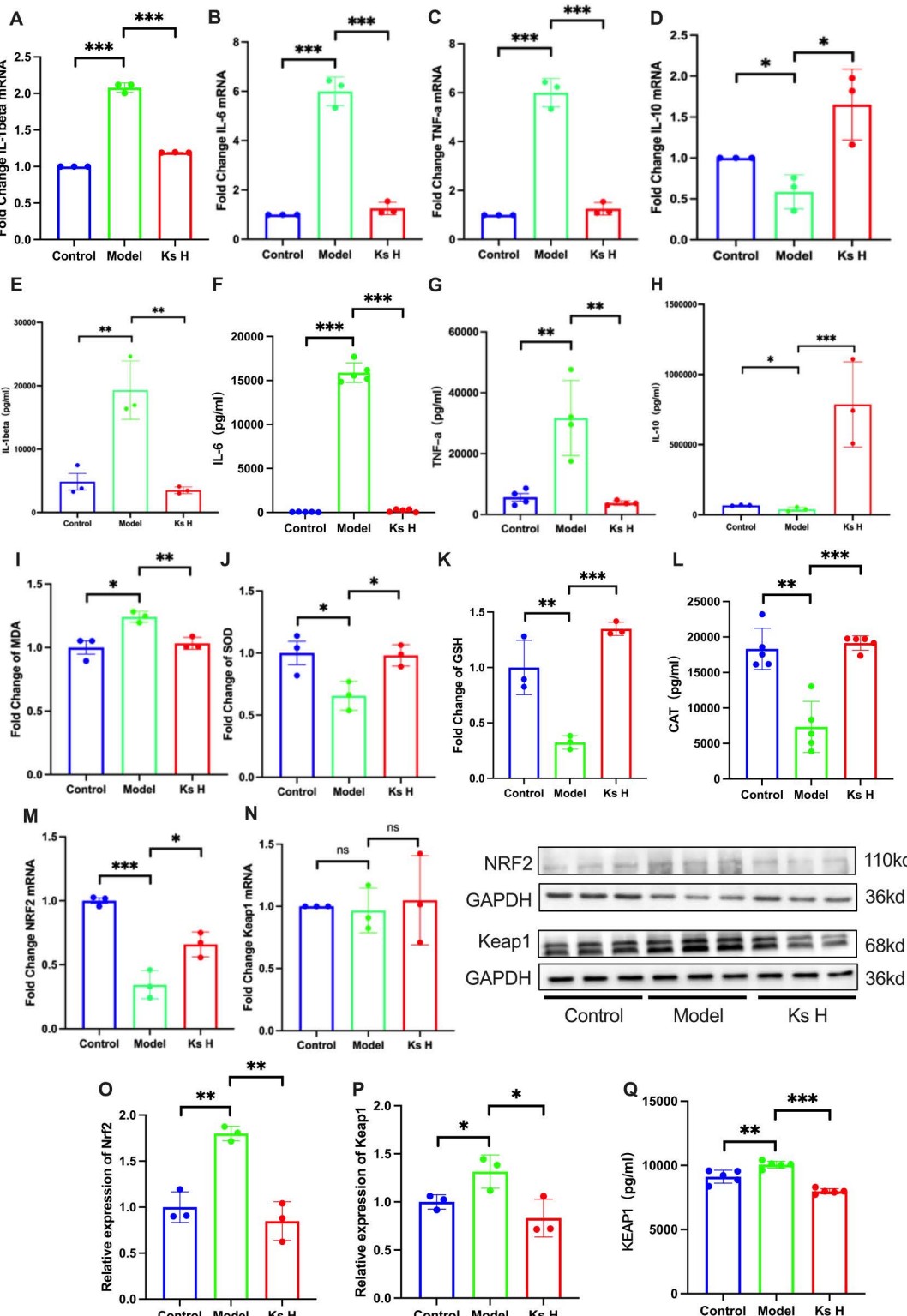

**Fig 3. Ks can alleviate oxidative stress and inflammation caused by APAP-induced liver injury and Ks promotes NRF2 nuclear translocation and activates the Nrf2/Keap1 pathway.** (A-D) Changes in the mRNA levels of pro-inflammatory cytokines (IL-1β, TNF-α, IL-6) and the

anti-inflammatory cytokine (IL-10).(D-H) Changes in the levels of pro-inflammatory cytokines (IL-1β, TNF-α, IL-6) and the anti-inflammatory cytokine (IL-10).(I-L) Changes in the levels of oxidative stress markers (MDA, GSH, SOD, CAT).(M, N) Changes in the mRNA levels of NRF2 and Keap1.(O, P) Results of protein blotting and densitometric analysis for NRF2 and Keap1.(Q) Changes in the protein levels of Keap1. *$P<0.05$, **$P<0.01$, ***$P<0.001$ indicate statistically significant differences compared with the control or model group.

changes observed in CerS6 and SPHK1 (S1 Fig I and J). As mentioned earlier, we performed network pharmacology by integrating the active compounds identified through mass spectrometry with the disease-related differential genes from the GSE241511 dataset. Notably, we observed that SPHK1, S1PR3, and S1PR5 in the sphingolipid metabolism pathway could be potential therapeutic targets for the prevention of acute liver injury induced by APAP, as mediated by the Ks extract. This finding has prompted us to focus more closely on the SPHK1/S1P/S1PR pathway (S2 FigA). Further analysis revealed that SPHK1 showed the most significant changes in sphingolipid metabolism-related genes in this dataset, and SPHK1, S1PR3, and S1PR5 are upstream and downstream regulators of S1P, a sphingolipid metabolite. Therefore, we assessed the expression of SPHK1, S1PR3, and S1PR5 using QPCR. The results showed that, at the mRNA level, SPHK1 was significantly upregulated following modeling, while it was notably reduced in the treatment group. However, no significant changes were observed for S1PR3 and S1PR5 between the APAP and Ks H groups (Fig 4H, O, and Q). Previous studies have shown that S1P is closely associated with alcoholic liver injury and cholestatic liver injury [34–36]. To investigate whether APAP-induced liver injury is associated with changes in S1P levels and whether S1P could serve as a potential drug target, we measured the serum S1P concentrations across different groups. The results showed that S1P levels were significantly increased in the model group, whereas they were markedly decreased in the treatment group (Fig 4I). Subsequently, we quantified the levels of two isoenzymes, SPHK1 and SPHK2, that regulate S1P synthesis using ELISA. The results indicated a significant increase in SPHK1 levels in the model group, which was significantly reduced in the treatment group. However, no significant changes in SPHK2 levels were observed in either the model or treatment groups (Fig 4J and K). Additionally, Western blot and immunohistochemistry results corroborated the changes in SPHK1 expression (Fig 4C and L). We further analyzed the mRNA expression of three downstream receptors of S1P using QPCR to explore the regulatory mechanisms of S1P. The results showed that changes in S1PR2 and S1PR4 were most significant. In the model group, the mRNA level of S1PR2 was markedly increased, which was significantly reduced upon treatment, while the trend for S1PR4 was the opposite of S1PR2 (Fig 4M, N, and P). ELISA-based protein quantification of S1PR2 and S1PR4 revealed that the protein expression of S1PR2 aligned with its mRNA levels, whereas the protein expression of S1PR4 exhibited an inverse trend compared to its mRNA levels (Fig 4R and S). These findings suggest that Ks effectively prevents APAP-induced acute liver injury in mice by regulating S1P and its upstream and downstream factors, including SPHK1, S1PR2, and S1PR4.

### 4.7. Molecular docking simulations of Ks with S1P, SPHK1, S1PR2, S1PR4, and NRF2-KEAP1

We performed molecular docking simulations of 13 major components of *Kummerowia striata* with target proteins using Autodock Vina 1.5.6. After completing the docking process, we selected the conformations that were most frequently repeated and exhibited the best binding affinities as the output results (Fig 5A). In the protein surface models, we primarily analyzed whether the ligand was located inside the protein or on its surface. As the docking was flexible, meaning the ligand conformation could change while the protein structure remained unchanged, we observed that the shape of the protein surface aligned well with the ligand's conformation. This suggests that the protein's natural structure facilitates small molecule binding. In the overall docking structure and 3D binding site analysis, we focused on the amino acid residues at the protein binding site that interacted with the ligand, as well as the number of hydrogen bonds. Generally, shorter hydrogen bonds correspond to lower energy and more stable systems. The typical hydrogen bond length is between 1.5–3.5 Å. In the 2D planar diagrams, we could clearly identify the amino acid residues at the binding site. It is commonly accepted

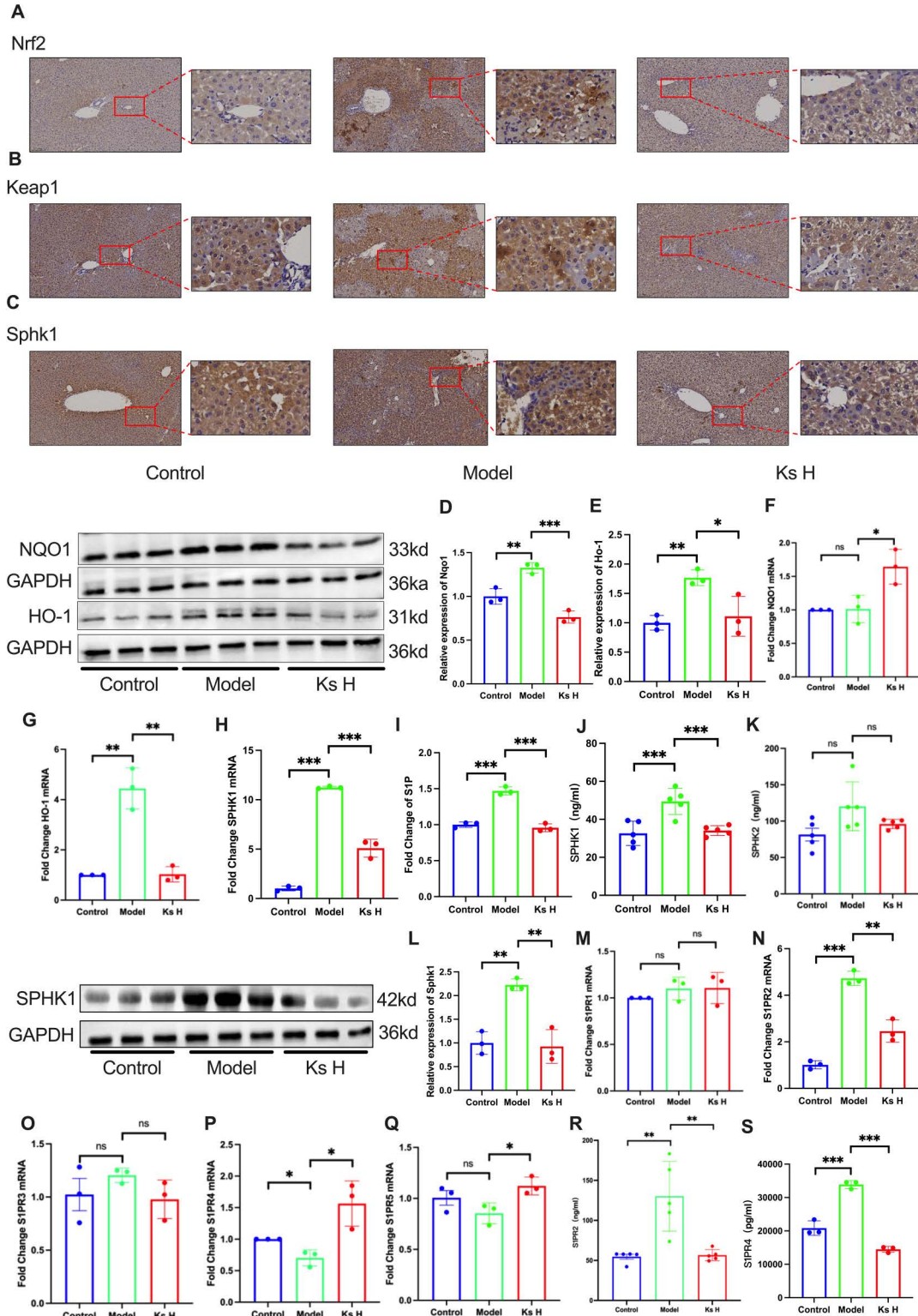

**Fig 4. Ks Inhibits the Metabolism of Sphingolipid Metabolite S1P and Improves APAP-Induced Acute Liver Injury.** (A-C) Immunohistochemical (IHC) staining of liver tissue for NRF2, Keap1, and SPHK1. (D, E) Western blot analysis of protein levels for NQO1 and HO-1 following NRF2 nuclear

translocation.(F, G) mRNA level changes of downstream targets NQO1 and HO-1 following NRF2 nuclear translocation.(H, J, L) Changes in mRNA levels and protein content of SPHK1, along with Western blot analysis of protein expression.(I) Changes in S1P levels.(K) Changes in SPHK1 protein levels.(M-Q) mRNA expression levels of S1PR1–5.(R, S) Changes in protein levels of S1PR2 and S1PR4.*$P < 0.05$, **$P < 0.01$, ***$P < 0.001$ indicate statistically significant differences compared with the control or model group.

that when the binding energy is less than −4 kcal/mol, the binding affinity is considered favorable, and smaller binding energies indicate stronger binding. We found that apigenin 7-O-glucuronide showed the best docking performance with SPHK1, S1PR2, and S1PR4, genistin exhibited the best docking performance with Nrf2-KEAP1, and stigmasterol had the best docking performance with lipid S1P, all of which had the lowest binding energies (Fig 5B–F).

### 4.8. Ks alleviates acetaminophen-induced liver injury in female mice

0As previously reported, we observed the protective effect of Ks against APAP-induced liver injury in male mice. To investigate whether Ks also exhibits hepatoprotective effects in females, we conducted an identical experiment with the same dosage in APAP-induced female mice. The levels of ALT, AST, LDH, inflammatory markers (IL-1β, IL-6, TNF-α), and oxidative stress indicators (SOD, CAT) in APAP-treated female mice exhibited similar trends to those observed in male mice. Furthermore, histopathological examination revealed that APAP also caused liver damage in female mice, and Ks was able to reverse this effect (Fig 6A~I). These findings provide strong evidence that Ks exerts hepatoprotective effects in female mice as well Fig 7.

## 5. Discussion

Traditional Chinese medicines (TCMs) have a natural advantage in the development of drugs for acute liver injury (AILI) due to their low side effects [37]. Currently, an increasing number of traditional TCMS monomers have been found to possess AILI effects, with 'silymarin', which has been widely applied in clinical practice, being one of them [38,39]. Silybin is the primary active component of silymarin, a complex mixture extracted from the seeds of the milk thistle plant. However, silymarin is contraindicated in patients allergic to members of the Asteraceae family and those with hormone-sensitive conditions. Additionally, it may induce adverse reactions such as headaches, diarrhea, nausea, bloating, musculoskeletal pain, loss of appetite, and fatigue [37]. Currently, the drugs primarily used in clinical practice for the treatment of AILI include NAC and silymarin. Therefore, the development of traditional Chinese medicines similar to silymarin holds significant importance [40,41]. Ks is a traditional herbal medicine commonly used in the Guangxi Zhuang Autonomous Region and is widely distributed across China. Research on Ks remains limited, with most studies focusing on renal disease models. Studies have shown that Ks possesses anti-inflammatory, antioxidant, and nephroprotective properties [42].

Nrf2 is a crucial transcription factor that regulates cellular ferroptosis and mitochondrial oxidative stress [20]. According to research, in the cytoplasm, Keap1 typically binds to Nrf2 and continuously degrades Nrf2 via the ubiquitin-proteasome pathway. Upon stimulation, Nrf2 can be released and translocate to the nucleus. Additionally, Nrf2 can alter the Keap1-NRF2 complex through competitive binding, leading to the dissociation of the complex and activation of the Nrf2 signaling pathway [21]. Several studies have confirmed that many drugs can improve mitochondrial morphology in cells with acute liver injury by upregulating Nrf2 and modulating its downstream factors, thereby inhibiting the occurrence of oxidative stress [22–24]. Previous studies have suggested that the expression of Keap1 in APAP induced liver injury models does not significantly differ from that in the control group, or that the expression of Keap1 is reduced after model induction [16,43]. However, in contrast, we found that the expression of Keap1 was significantly increased in the model group, and that Ks was able to inhibit the expression of Keap1. We hypothesize that the increase in Keap1 in the model group promoted the stability of Nrf2 in the cytoplasm, thereby hindering its nuclear translocation. This resistance to oxidative stress induced by APAP exacerbated the occurrence of oxidative stress and led to a compensatory upregulation of Nrf2

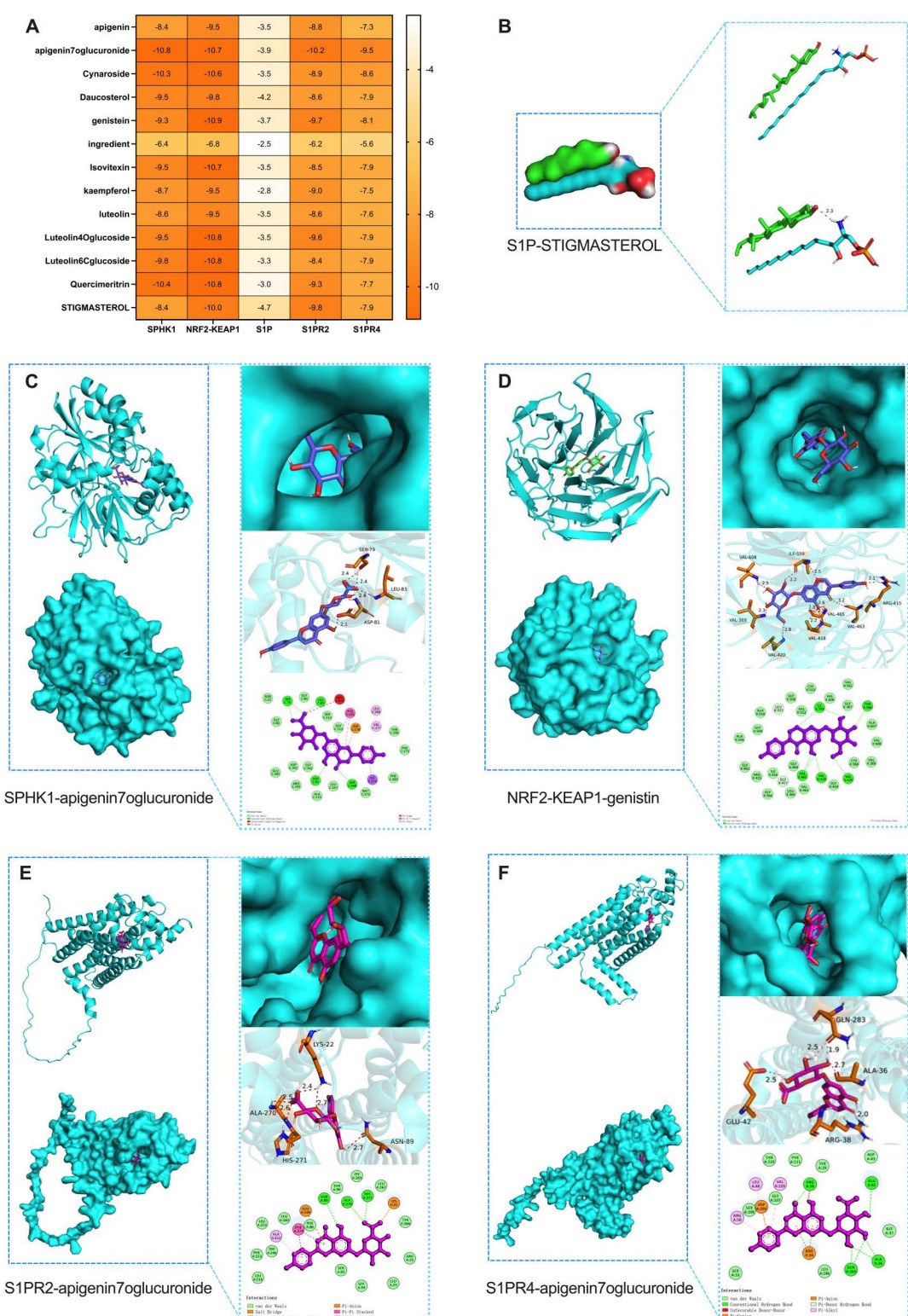

**Fig 5. Molecular Docking Simulations of Ks with SPHK1, S1PR2, S1PR4, and NRF2-KEAP1.** (A) A heatmap of the binding affinity of the 13 components of Ks with S1P, SPHK1, S1PR2, S1PR4, and NRF2-KEAP1, showing the best docking results. (B) Docking results of S1P with stigmasterol.(C) Docking results of SPHK1 with apigenin 7-O-glucuronide.(D) Docking results of NRF2-KEAP1 with genistin. (E) Docking results of S1PR2 with apigenin 7-O-glucuronide.(F) Docking results of S1PR4 with apigenin 7-O-glucuronide. (Specific analysis can be found in the original text.).

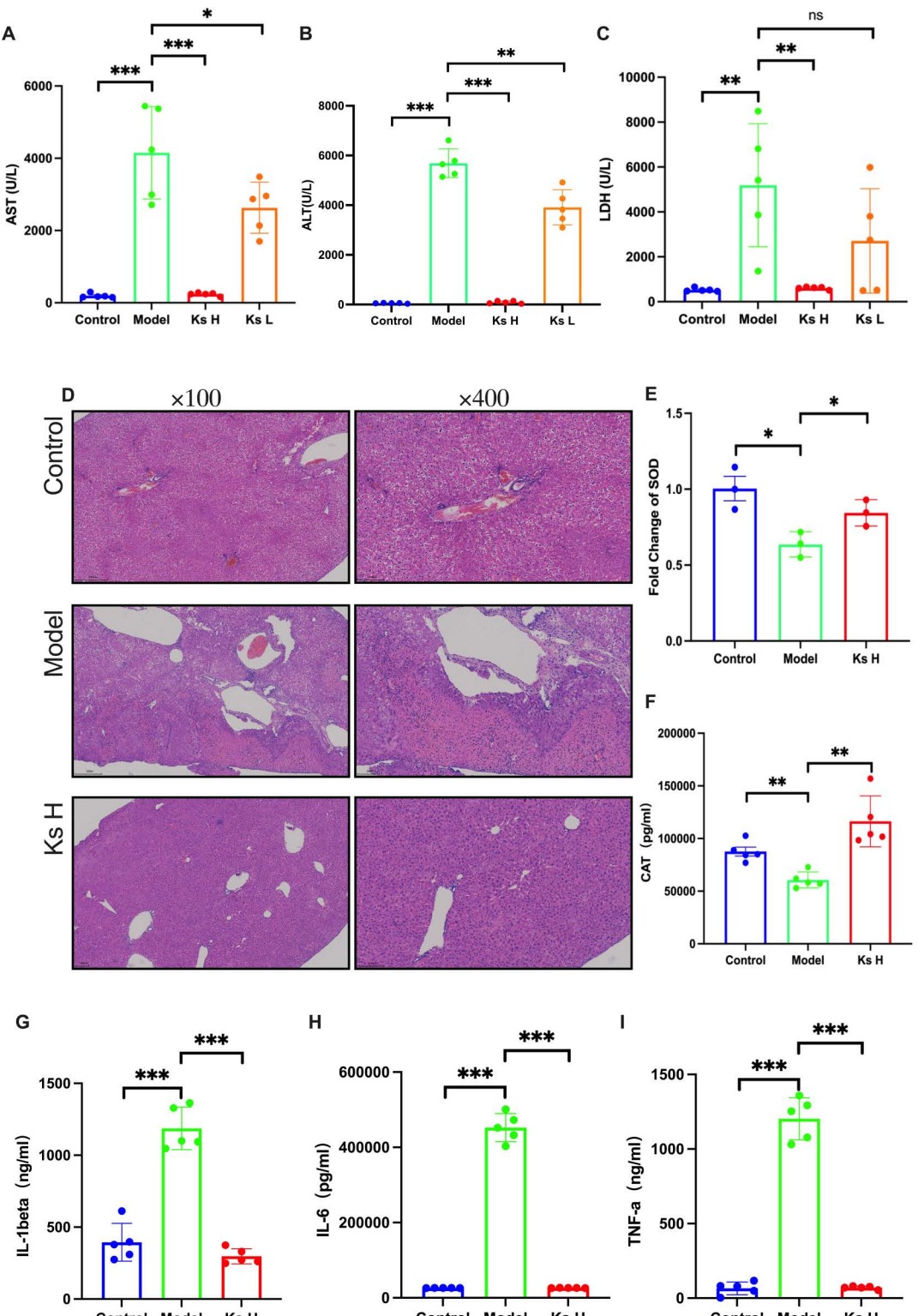

**Fig 6. Ks alleviates acetaminophen-induced liver injury in female mice.** (A-C) Serum levels of AST, ALT, and LDH were measured to assess liver function.(D) H&E staining of liver tissue sections.(E,F) Changes in oxidative stress markers (SOD, CAT) levels. (G-I) Changes in the levels of anti-inflammatory factors (IL-1β, IL-6, TNF-α). *P<0.05, **P<0.01, ***P<0.001 indicate statistically significant differences compared with the control or model group.

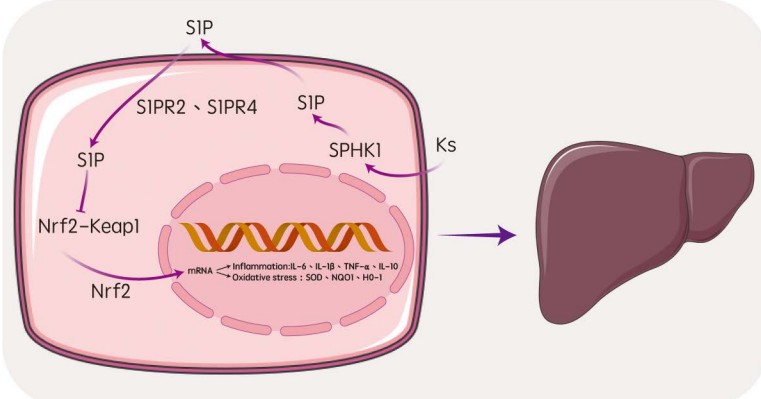

**Fig 7. Molecular Mechanisms of Ks in the Prevention of APAP-Induced Acute Liver Injury.**

expression. The compensatory upregulation of Nrf2 triggered the expression of its downstream antioxidant factors, NQO1 and HO-1, in an attempt to combat oxidative stress and inflammation.

Recent studies have also highlighted the role of the TNF signaling pathway in drug-induced liver injury, particularly in mediating inflammation and hepatocyte apoptosis. TNF-α, a pro-inflammatory cytokine, has been shown to promote the progression of APAP-induced liver injury by activating downstream signaling cascades such as NF-κB and JNK, further exacerbating oxidative stress and inflammation. Although this study did not directly assess TNF-α or its associated pathways, the observed reduction in hepatic inflammation following Ks treatment suggests a potential modulatory effect on TNF-related signaling. Future studies will be necessary to investigate whether the protective effects of Ks are mediated, at least in part, through suppression of the TNF pathway, thereby providing additional insight into its anti-inflammatory mechanisms.

Meanwhile, Ks, by inhibiting the expression of Keap1, alleviated the occurrence of oxidative stress, resulting in the restoration of Nrf2 and its downstream antioxidant genes to normal levels in the Ks H group. It is important to note that since we collected samples 24 hours post-modeling, a time point considered to be when liver injury is most severe, liver repair begins after 24 hours, and compensatory effects gradually subside. Moreover, gene expression typically occurs prior to changes at the protein level. Therefore, the inhibition of Keap1 expression by Ks reduces the degradation of Nrf2, which leads to a decrease in Nrf2 gene expression in the model group and an increase after administration. However, we believe that Ks does not directly affect the expression of Keap1 but rather regulates its expression indirectly through other pathways. Molecular docking results suggest that Ks may modulate the interaction between Nrf2 and Keap1, thereby influencing the expression and interaction of both, supporting our hypothesis.

Furthermore, upon reviewing the literature, we found that although most studies suggest that Keap1 and Nrf2 should exhibit opposing trends, or that Nrf2 expression should be decreased in liver injury, recent research in other diseases has shown that the concurrent depletion of Keap1 and Nrf2 can rescue cells from death [44]. This cell death mechanism, primarily regulated by changes in Keap1, is termed "Oxeiptosis". Currently, there is no research on Oxeiptosis in liver injury or liver diseases, which is why we have not conducted further mechanistic studies [45]. Oxeiptosis is primarily mediated by reactive oxygen species (ROS), and the pathogenesis of many liver diseases, including liver injury, is closely associated with ROS-mediated oxidative stress and inflammation. Based on our experimental findings, it may be suggested that Oxeiptosis could be one of the potential directions for future research in the treatment of liver diseases.

In our previous studies, we found that genes associated with sphingolipid metabolism, such as CERK and B4GALT6, were upregulated in the liver of APAP-induced acute liver injury model. Ks was able to inhibit the expression of CERK but

did not affect the expression of B4GALT6 (S1 FigE and F). IHC results further validated the changes in CERK at the protein level. However, due to the lack of significant inhibition by Ks, we did not conduct further studies (S1 FigH). Previous studies have shown that silencing CERK can inhibit the expression of Nrf2, suggesting that Ks may also influence Nrf2 expression through the regulation of CERK gene expression.

S1P is one of the key bioactive sphingolipid metabolites, playing crucial roles in processes such as cell proliferation, apoptosis, and inflammation, and is thus referred to as a "bioactive sphingolipid'' [19]. The levels of S1P are regulated by two isoenzymes, SPHK1 and SPHK2 [18]. Previous studies have demonstrated that mice with knockout of the SPHK1 gene exhibit significant resistance to liver injury induced by APAP, whereas no related studies have been reported for SPHK2 [17]. We found that Ks can inhibit the effect of SPHK1 in regulating S1P levels, but no significant differences were observed for SPHK2 across different groups. Moreover, S1P exerts its effects by binding to G-protein coupled receptors, namely S1PR1–5, with recent research primarily focusing on S1PR2. Among all studies on liver injury, only S1PR1 and S1PR3 have shown significant changes. For the first time, we observed that in an APAP-induced liver injury model, the expression of S1PR2 and S1PR4 was significantly upregulated, and Ks was able to inhibit the expression of these receptors at the protein level. Although inconsistent changes in S1PR4 were detected at both mRNA and protein levels, we believe this may be related to transcriptional and post-transcriptional modifications, which warrants further investigation.

Previous studies have generally suggested that liver injury occurs more frequently and severely in males, primarily due to the influence of estrogen on liver injury. Therefore, much of the research on liver injury mechanisms and hepatoprotective drugs has focused on male mice, resulting in a relative lack of studies on female mice. However, we note that some studies indicate a higher incidence and severity of DILI in females, particularly postmenopausal women. This has prompted us to explore whether Ks exerts a similar hepatoprotective effect in female mice, as observed in male mice [46]. We observed that under the same dose of APAP treatment, the extent of liver injury in female mice was significantly lower than that in male mice. However, despite this, female mice still developed acute liver injury, which was significantly alleviated by Ks. Nevertheless, whether the mechanism of action of Ks in female mice is the same as in male mice requires further investigation. Additionally, we conducted experiments in both male and female KM mice to assess the preventive effects of Ks on carbon tetrachloride (CCl4)-induced liver injury. The doses and procedures used in male KM mice were consistent with those in the APAP experiment (Fig 1A), while in female KM mice, only high-dose gavage treatment was performed. Elevated serum levels of alanine aminotransferase(ALT), aspartate aminotransferase (AST), and (lactate dehydrogenase) LDH reflect hepatocellular damage and necrosis. Therefore, monitoring these enzymes provides important evidence to assess the extent of liver injury and the hepatoprotective effect of Ks treatment in our APAP-induced liver injury model. In our study, the results of ALT and LDH assays further confirmed that Ks could prevent liver injury induced by CCl4 in both male and female KM mice (Supplementary Fig 1 A–D in Supporting information).

In summary, we not only identified and validated the hepatoprotective effects of Ks, but also revealed a novel mechanism involving S1P/S1PR2/S1PR4 signaling in APAP-induced acute liver injury in mice. We hypothesize that Ks extracts may exert protective effects against APAP-induced acute liver injury by modulating the SPHK1/S1P/S1PR2/S1PR4 pathway and the Nrf2/Keap1 pathway, thereby mediating oxidative stress and inflammatory responses.

## Supporting information

**S1 Fig. *Ks ameliorates* carbon tetrachloride (CCl$_4$)induced liver injury and identifies sphingolipid-related targets in APAP-treated mice (A-B) Changes in serum ALT and LDH levels in male Km mice with acute liver injury induced by carbon tetrachloride (CCl$_4$) following Ks treatment** (C-D) Changes in serum ALT and LDH levels in female Km mice with acute liver injury induced by CCl$_4$ following Ks treatment.(E, H) mRNA expression of CERK in liver tissues of male C57/B6J mice with acute liver injury induced by APAP, and IHC staining images.(F) mRNA expression of B4GALT6

in liver tissues of male C57/B6J mice with acute liver injury induced by APAP following Ks treatment.(I,J)Waterfall plot of the most significantly altered sphingolipid metabolic pathway genes (SPHK1, Cer S6) in dataset GSE241511.
(JPG)

**S2 Fig. The network pharmacology model was constructed by integrating the active components of the Ks extract identified via UPLC-Q-TOF-MS analysis with genes that significantly changed after APAP modeling, based on the GSE241511 dataset.** (A) Drug Compound Target Pathway-Disease Network. The depth of color represents the strength of targeting effect. (B) Venn Diagram of Targets of Ks Associated with Acute Liver Injury (AILI).(C) Interaction of Key Genes.(D) GO Enrichment Analysis of Target Genes.(E) KEGG Pathway Analysis of Target Genes. The bubble size indicates the number of genes in the pathway, and the color represents the p-value.
(TIFF)

**S3 Fig. UPLC-Q-TOF-MS analysis of Ks extract.**
(JPG)

**S1 Raw_images. All orginal westernblot images.**
(ZIP)

## Author contributions

**Conceptualization:** Youlan Xie, Hang liu, Yubo Xiao, Lanyu Li.

**Data curation:** Lanyu Li.

**Formal analysis:** Lanyu Li.

**Methodology:** Huanghui Qin, Youlan Xie, Runxiao Chen.

**Validation:** Hang liu.

**Writing – original draft:** Huanghui Qin, Hang liu.

**Writing – review & editing:** Yubo Xiao.

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
