## [Decision Letter · Decision Letter 0]

20 May 2025

Dear Dr. Li,

Thank you for submitting your manuscript to PLOS ONE. After careful consideration, we feel that it has merit but does not fully meet PLOS ONE’s publication criteria as it currently stands. Therefore, we invite you to submit a revised version of the manuscript that addresses the points raised during the review process.

Thank you for choosing our journal for submission of your research work. Please address all Reviewer's and Editor's comments and amend your manuscript accordingly. 

A rebuttal letter that responds to each point raised by the academic editor and reviewer(s). You should upload this letter as a separate file labelled 'Response to Reviewers'.A marked-up copy of your manuscript that highlights changes made to the original version. You should upload this as a separate file labeled 'Revised Manuscript with Track Changes'.An unmarked version of your revised paper without tracked changes. You should upload this as a separate file labeled 'Manuscript'.

We look forward to receiving your revised manuscript.

Kind regards,

Olga A Sukocheva, PhD

Academic Editor

PLOS ONE

“This research was supported by Guangxi Natural Science Foundation Joint Special Project (Guilin Medical University Special Project, 2024JJH130036) and Natural Science Foundation of Hunan Province of China (2023JJ50440).”

“This research was supported by Guangxi Natural Science Foundation Joint Special Project (Guilin Medical University Special Project, 2024JJH130036) and Natural Science Foundation of Hunan Province of China (2023JJ50440).”

“This research was supported by Guangxi Natural Science Foundation Joint Special Project (Guilin Medical University Special Project, 2024JJH130036) and Natural Science Foundation of Hunan Province of China (2023JJ50440).”

**Additional Editor Comments:**

The study tested preventive effects of Kummerowia striata (Ks) (extracts? - indicate this and the extract content) against Acetaminophen (APAP)-induced liver injury in vivo, using a mouse model. Ks contains compounds with anti-inflammatory and antioxidant activities. Authors reports that Ks attenuated acute liver injury via S1P signalling pathway.

The study is interesting and novel, although it should be amended.

1. Abstract: there is no need to introduce abbreviation DILI in the Abstract as you use it only once. However, S1P and SphK/S1P4 abbreviations were not introduced at all and should be deciphered properly in the Abstract.

2. Abstract: Discussion section is missing.

3. Keywords should include sphingolipids and sphingosine kinase

4. Introduction: All abbreviations should be deciphered. SPHK – sphingosine kinase, S1P -sphingosine-1-phospahte etc…It is necessary to introduce sphingolipid Signaling pathway in more details. It would be better to cite few recent large reviews in this area. I suggest checking these manuscripts https://pubmed.ncbi.nlm.nih.gov/31863815/ and https://pubmed.ncbi.nlm.nih.gov/38698424/

5. Methods section should contain citations/references for the used methods.

6. IHC images should include higher resolution inserts focused on the cellular structure. Figure 4 A is of poor quality . please provide better images/higher quality/contrasted.

7. Discussion section should include the discussion of the potential link to the pro-inflammatory processes including TNF-signalling pathway.

Reviewers' comments:

Reviewer's Responses to Questions

**Comments to the Author**

1. Is the manuscript technically sound, and do the data support the conclusions?

Reviewer #1: Yes

2. Has the statistical analysis been performed appropriately and rigorously?

Reviewer #1: Yes

3. Have the authors made all data underlying the findings in their manuscript fully available?

Reviewer #1: Yes

4. Is the manuscript presented in an intelligible fashion and written in standard English?

Reviewer #1: Yes

Reviewer #1: The manuscript "Kummerowia striata extract protects paracetamol-induced liver injury by modulating the S1P/Nrf2/Keap1 pathway" by Huanghui Qin et al. is devoted to the study of the protective effect of Kummerowia striata against acute liver injury caused by acetaminophen and further study of its potential molecular mechanisms.

The degree of relevance of the article provided is beyond doubt, since drug-induced liver damage is a serious problem both in the development of therapeutic agents and in the clinical use of the drug.

The manuscript describes the results obtained using modern methods and will be of interest to the scientific community. The design of the study corresponds to the goals set by the authors.

In my opinion, the manuscript can be accepted for publication after revision.

1.Regarding the "Abstract" section. Based on the title of the manuscript and the text, it is obvious that we are talking about the extract of Kummerowia striata, and not about the plant itself. However, when describing the subsection "Methods" in the abstract, the authors present the information in such a way that the animals receive the plant directly, and not its extract. This drawback should be eliminated.

2. As for the "Introduction" section, this section does not contain comprehensive information, looks disorganized and requires significant revision. The authors need to supplement and amend the following provisions:

- Due to the fact that the authors of the manuscript aim to study the SPHK1/S1P signaling pathway as a therapeutic target, it is necessary to add the information about SPHK1/S1P. Currently, based on the "Introduction" section, the choice of this signaling pathway is not obvious.

- The authors should review the text fragment describing Kummerowia striata. It is necessary to reveal the fact why this particular plant was chosen. Currently, the authors have described the therapeutic potential of this plant very succinctly. It makes it absolutely impossible to verify the expediency of using Kummerowia striata as an object of research. Moreover, the paragraph on Kummerowia striata needs additional information on the therapeutic properties of the plant, indicating additional sources of literature (it is unacceptable to cite only one link to the article). Authors should not neglect to cite literary sources. Currently, it seems that most of the data is the authors' own discovery, but this is wrong.

- The authors also did not pay enough attention to Nrf2. Why was Nrf2 chosen as a potential target? Similarly to the above remarks, the text of the manuscript lacks justification for its choice. How is this factor related to drug-induced liver damage? Currently, the only sentence about Nrf2 at the very end of the Introduction section looks out of place.

3. The authors neglected to decipher the abbreviations. It is necessary to check the entire text and enter the transcripts at the first mention. For example, it is not obvious what SPHK1 is. A similar problem is present for most other abbreviations.

4. The authors should describe the methodological part of the manuscript in more detail. In particular, the authors neglected the detailed description of the molecular docking procedure.

5. For what purpose did the authors study the levels of ALT, AST, and LDH? It is necessary to disclose the expediency of studying them in the "Discussion" section.

6. After describing the method in paragraph 2.10, the authors repeat the information about the blotting procedure - it is necessary to delete this text fragment.

7. Figure 1A does not fully describe the experimental scheme. This diagram is more like a description of the drug administration to only 3 of the 6 groups.

8. What does Figure 7 relate to? Why did the authors include it if it is not mentioned in the text of the manuscript?

9. I would also like to receive an answer to the following questions:

- Have studies of Kummerowia striata extract been conducted on cell cultures in order to obtain information about its cytotoxic profile? If not, has this information been searched in the literature? Please briefly report these results, because without this information it is unacceptable to conduct experiments on animals.

- What further prospects do the authors see for this work? Will the authors identify the individual components of the Kummerowia striata extract, or are they planning to use the extract as a ready-made medicinal product?

**Do you want your identity to be public for this peer review?** For information about this choice, including consent withdrawal, please see our Privacy Policy

Reviewer #1: No

---

## [Author Response · Author response to Decision Letter 1]

29 Jun 2025

Additional Editor Comments:

The study tested preventive effects of Kummerowia striata (Ks) (extracts? - indicate this and the extract content) against Acetaminophen (APAP)-induced liver injury in vivo, using a mouse model. Ks contains compounds with anti-inflammatory and antioxidant activities. Authors reports that Ks attenuated acute liver injury via S1P signalling pathway.

The study is interesting and novel, although it should be amended.

Reviewer Comment 1:Abstract: there is no need to introduce abbreviation DILI in the Abstract as you use it only once. However, S1P and SphK/S1P4 abbreviations were not introduced at all and should be deciphered properly in the Abstract.

We thank the reviewer for this valuable comment. In the revised manuscript, we have removed the abbreviation “DILI” from the Abstract, as it appears only once. Additionally, we have properly introduced and defined the abbreviations “S1P” (sphingosine-1-phosphate) and “SPHK” (sphingosine kinase), along with their relevant receptor “S1PR4” (sphingosine-1-phosphate receptor 4) at their first mention in the Abstract to enhance clarity for readers.

Reviewer Comment 2: Abstract: Discussion section is missing.

Thank you for the suggestion. We have now included a brief discussion at the end of the Abstract, summarizing the main findings and highlighting the potential mechanisms of Ks in attenuating APAP-induced liver injury via regulation of oxidative stress and the sphingolipid signaling pathway.

Reviewer Comment 3:Keywords should include sphingolipids and sphingosine kinase.

We appreciate the reviewer’s suggestion. We have revised the Keywords section to include both “sphingolipids” and “sphingosine kinase” to better reflect the study’s scope and facilitate indexing.

Reviewer Comment 4: Introduction: All abbreviations should be deciphered. SPHK – sphingosine kinase, S1P -sphingosine-1-phospahte etc…It is necessary to introduce sphingolipid Signaling pathway in more details. It would be better to cite few recent large reviews in this area. I suggest checking these manuscripts https://pubmed.ncbi.nlm.nih.gov/31863815/ and https://pubmed.ncbi.nlm.nih.gov/38698424/

We sincerely thank the reviewer for this helpful suggestion. In the revised manuscript, we have carefully reviewed the entire text and provided full definitions for all abbreviations upon their first appearance, including sphingosine kinase (SPHK) and sphingosine-1-phosphate (S1P), among others. In addition, we have significantly expanded the introduction to include a more comprehensive description of the sphingolipid signaling pathway, highlighting its relevance to oxidative stress, inflammation, and liver injury. We have also incorporated citations of the two recent review articles recommended by the reviewer (PMID: 31863815 and 38698424) to support this section and provide readers with updated background knowledge on the topic.

Reviewer Comment 5: Methods section should contain citations/references for the used methods.

We thank the reviewer for pointing this out. In the revised manuscript, we have carefully reviewed the Methods section and added appropriate citations for all key techniques used in the study. For example, the 3D protein structures used in molecular docking were obtained from the RCSB Protein Data Bank, and we have now cited the corresponding reference:Burley SK, Bhatt R, Bhikadiya C, et al. Updated resources for exploring experimentally-determined PDB structures and Computed Structure Models at the RCSB Protein Data Bank. Nucleic Acids Res. 2025;53(D1):D564–D574. doi:10.1093/nar/gkae1091. Furthermore, molecular docking was performed using AutoDock Vina 1.5.6, and we have now cited:Trott O, Olson AJ. AutoDock Vina: improving the speed and accuracy of docking with a new scoring function, efficient optimization, and multithreading. J Comput Chem. 2010;31(2):455–461.

Reviewer Comment 6: IHC images should include higher resolution inserts focused on the cellular structure. Figure 4 A is of poor quality. please provide better images/higher quality/contrasted.

We thank the reviewer for this constructive comment. In response, we have replaced Figure 4A with a higher-resolution version that includes improved contrast and clearer visualization of the cellular structures. The updated image has been uploaded as a revised figure file in the attachment, as required. We believe this change significantly enhances the clarity and interpretability of the immunohistochemical results.

Reviewer Comment 7: Discussion section should include the discussion of the potential link to the pro-inflammatory processes including TNF-signalling pathway.

We thank the reviewer for this constructive comment. Recent studies have highlighted the critical role of the tumor necrosis factor (TNF) signaling pathway in drug-induced liver injury, particularly in promoting inflammation and hepatocyte apoptosis. TNF-α, a key pro-inflammatory cytokine, has been shown to aggravate APAP-induced hepatic injury by activating downstream pathways such as NF-κB and JNK, thereby intensifying oxidative stress and inflammatory responses. Although our current study did not directly examine TNF-α or its related signaling cascades, the observed attenuation of liver inflammation following Ks treatment suggests a possible modulatory effect on TNF-associated pathways. Further investigation will be necessary to determine whether the hepatoprotective effects of Ks are partially mediated through the inhibition of TNF signaling, which would provide additional insight into its anti-inflammatory mechanisms.

Review Comments to the Author:

Reviewer #1: The manuscript "Kummerowia striata extract protects paracetamol-induced liver injury by modulating the S1P/Nrf2/Keap1 pathway" by Huanghui Qin et al. is devoted to the study of the protective effect of Kummerowia striata against acute liver injury caused by acetaminophen and further study of its potential molecular mechanisms.

The degree of relevance of the article provided is beyond doubt, since drug-induced liver damage is a serious problem both in the development of therapeutic agents and in the clinical use of the drug.

The manuscript describes the results obtained using modern methods and will be of interest to the scientific community. The design of the study corresponds to the goals set by the authors.In my opinion, the manuscript can be accepted for publication after revision.

1. Regarding the "Abstract" section. Based on the title of the manuscript and the text, it is obvious that we are talking about the extract of Kummerowia striata, and not about the plant itself. However, when describing the subsection "Methods" in the abstract, the authors present the information in such a way that the animals receive the plant directly, and not its extract. This drawback should be eliminated.

We thank the reviewer for pointing out this important issue. In the revised manuscript, we have corrected the description in the "Methods" subsection of the Abstract to clearly indicate that the animals were treated with Kummerowia striata extract, rather than the raw plant material. We believe this revision eliminates the ambiguity and aligns the Abstract with the actual experimental procedures.

2. As for the "Introduction" section, this section does not contain comprehensive information, looks disorganized and requires significant revision. The authors need to supplement and amend the following provisions:

- Due to the fact that the authors of the manuscript aim to study the SPHK1/S1P signaling pathway as a therapeutic target, it is necessary to add the information about SPHK1/S1P. Currently, based on the "Introduction" section, the choice of this signaling pathway is not obvious.

-the authors should review the text fragment describing Kummerowia striata. It is necessary to reveal the fact why this particular plant was chosen. Currently, the authors have described the therapeutic potential of this plant very succinctly. It makes it absolutely impossible to verify the expediency of using Kummerowia striata as an object of research. Moreover, the paragraph on Kummerowia striata needs additional information on the therapeutic properties of the plant, indicating additional sources of literature (it is unacceptable to cite only one link to the article). Authors should not neglect to cite literary sources. Currently, it seems that most of the data is the authors' own discovery, but this is wrong.

- The authors also did not pay enough attention to Nrf2. Why was Nrf2 chosen as a potential target? Similarly to the above remarks, the text of the manuscript lacks justification for its choice. How is this factor related to drug-induced liver damage? Currently, the only sentence about Nrf2 at the very end of the Introduction section looks out of place.

We thank the reviewer for pointing out this important issue. The introduction section has been supplemented and revised in the text.

3. The authors neglected to decipher the abbreviations. It is necessary to check the entire text and enter the transcripts at the first mention. For example, it is not obvious what SPHK1 is. A similar problem is present for most other abbreviations.

We sincerely thank the reviewer for this helpful suggestion. In the revised manuscript, we have carefully reviewed the entire text and provided full definitions for all abbreviations upon their first appearance, including sphingosine kinase (SPHK) and sphingosine-1-phosphate (S1P), among others.

4. The authors should describe the methodological part of the manuscript in more detail. In particular, the authors neglected the detailed description of the molecular docking procedure.

We thank the reviewer for this valuable suggestion. In the revised manuscript, we have supplemented the methodological section with additional details regarding the molecular docking procedure. Specifically, the 3D protein structure of the target compound was obtained from the Protein Data Bank (PDB). The 2D chemical structure of the small molecule was downloaded from PubChem, a comprehensive chemical compound repository maintained by the National Center for Biotechnology Information (NCBI). Molecular docking was performed using AutoDock Vina 1.5.6. After docking, the conformation with the highest frequency and best binding affinity was selected as the final result.

5. For what purpose did the authors study the levels of ALT, AST, and LDH? It is necessary to disclose the expediency of studying them in the "Discussion" section.

We thank the reviewer for this valuable suggestion. Elevated serum levels of alanine aminotransferase (ALT), aspartate aminotransferase (AST), and lactate dehydrogenase (LDH) are widely recognized indicators of hepatocellular damage and necrosis. Therefore, monitoring these enzyme levels provides critical evidence for assessing the degree of liver injury and evaluating the hepatoprotective effects of Ks treatment in our APAP-induced liver injury model. In the revised manuscript, we have added this explanation to the Discussion section. Moreover, in our study, the results of ALT and LDH assays further confirmed that Ks could also prevent CCl₄-induced liver injury in both male and female KM mice (Supplementary Fig. A–D), reinforcing its protective role across different models of liver damage.

6. After describing the method in paragraph 2.10, the authors repeat the information about the blotting procedure - it is necessary to delete this text fragment.

We thank the reviewer for the helpful observation. The redundant description of the blotting procedure following paragraph 2.10 has been carefully removed to avoid unnecessary repetition and improve the clarity of the Methods section.

7. Figure 1A does not fully describe the experimental scheme. This diagram is more like a description of the drug administration to only 3 of the 6 groups.

We thank the reviewer for pointing out this important issue. The article has has been fully describe the experimental scheme in Figure 1A.

8. What does Figure 7 relate to? Why did the authors include it if it is not mentioned in the text of the manuscript?

Thank you for the helpful comment. Figure 7 is a graphical abstract intended to illustrate the proposed mechanism by which Kummerowia striata (Ks) exerts its hepatoprotective effects through modulation of the SPHK1/S1P signaling pathway. We apologize for the oversight of not referencing it in the main text. This figure has now been revised and uploaded as a standalone graphical abstract to provide a visual summary of the study’s findings.

9. I would also like to receive an answer to the following questions:

- Have studies of Kummerowia striata extract been conducted on cell cultures in order to obtain information about its cytotoxic profile? If not, has this information been searched in the literature? Please briefly report these results, because without this information it is unacceptable to conduct experiments on animals.

Thank you for your important question. Prior to conducting animal experiments, we performed a cytotoxicity assay using the CCK-8 method on AML-12 cells. Ks extract was tested at concentrations ranging from 10 μg/mL to 1280 μg/mL, as following figure. The results indicated that none of the tested concentrations caused significant cytotoxicity compared with the control group. This suggests that Ks extract has a favorable safety profile in vitro and supports its potential for further investigation as a therapeutic agent for liver injury.

- What further prospects do the authors see for this work? Will the authors identify the individual components of the Kummerowia striata extract, or are they planning to use the extract as a ready-made medicinal product?

Thank you for the insightful question. In future studies, we plan to further isolate and characterize the active components of the Kummerowia striata (Ks) extract using chromatographic and spectrometric methods such as UPLC-MS/MS. Identifying the key bioactive compounds will allow us to clarify their mechanisms of action and evaluate their individual therapeutic potential. Ultimately, our goal is to determine whether a standardized extract or specific purified compound(s) from Ks would be more suitable for drug development. Both approaches—using the extract as a whole and isolating active components—will be explored in parallel to maximize translational potential.

---

## [Decision Letter · Decision Letter 1]

23 Jul 2025

Kummerowia striata extract protects paracetamol-induced liver injury by modulating the S1P/Nrf2/Keap1 pathway

PONE-D-25-21418R1

Dear Dr. Li,

We’re pleased to inform you that your manuscript has been judged scientifically suitable for publication and will be formally accepted for publication once it meets all outstanding technical requirements.

Kind regards,

Olga A Sukocheva, PhD

Academic Editor

PLOS ONE

Additional Editor Comments (optional):

Thank you for addressing all comments and suggestions properly.

Reviewers' comments:

Reviewer's Responses to Questions

**Comments to the Author**

Reviewer #1: All comments have been addressed

Reviewer #2: All comments have been addressed

2. Is the manuscript technically sound, and do the data support the conclusions?

Reviewer #1: Yes

Reviewer #2: Yes

3. Has the statistical analysis been performed appropriately and rigorously?

Reviewer #1: Yes

Reviewer #2: Yes

4. Have the authors made all data underlying the findings in their manuscript fully available?

Reviewer #1: Yes

Reviewer #2: Yes

5. Is the manuscript presented in an intelligible fashion and written in standard English?

Reviewer #1: Yes

Reviewer #2: Yes

Reviewer #1: I would like to thank the team of highly respected authors for their work on improving the manuscript.

Currently, the manuscript looks like a well-organized and completed study and can be published.

Reviewer #2: All reviewers' comment had been addressed properly. The revised manuscript can be accepted. Authors amended the manuscript to higher level.

**Do you want your identity to be public for this peer review?** For information about this choice, including consent withdrawal, please see our Privacy Policy

Reviewer #1: No

Reviewer #2: No

---

## [Editor Report · Acceptance letter]

PONE-D-25-21418R1

PLOS ONE

Dear Dr. Li,

I'm pleased to inform you that your manuscript has been deemed suitable for publication in PLOS ONE. Congratulations! Your manuscript is now being handed over to our production team.

Kind regards,

on behalf of

Dr. Olga A Sukocheva

Academic Editor

PLOS ONE